# SAR-HUB: Pre-Training, Fine-Tuning, and Explaining

Haodong Yang [1], Xinyue Kang [2], Long Liu [1], Yujiang Liu [1] and Zhongling Huang [1,*]

1    The BRain and Artificial INtelligence Lab (BRAIN LAB), School of Automation,
     Northwestern Polytechnical University, Xi'an 710072, China; hdyang@mail.nwpu.edu.cn (H.Y.);
     liulong2019@mail.nwpu.edu.cn (L.L.); yujiang@mail.nwpu.edu.cn (Y.L.)
2    School of Civil Aviation, Northwestern Polytechnical University, Xi'an 710072, China; kxy@mail.nwpu.edu.cn
*    Correspondence: huangzhongling@nwpu.edu.cn

**Abstract:** Since the current remote sensing pre-trained models trained on optical images are not as effective when applied to SAR image tasks, it is crucial to create sensor-specific SAR models with generalized feature representations and to demonstrate with evidence the limitations of optical pre-trained models in downstream SAR tasks. The following aspects are the focus of this study: pre-training, fine-tuning, and explaining. First, we collect the current large-scale open-source SAR scene image classification datasets to pre-train a series of deep neural networks, including convolutional neural networks (CNNs) and vision transformers (ViT). A novel dynamic range adaptive enhancement method and a mini-batch class-balanced loss are proposed to tackle the challenges in SAR scene image classification. Second, the pre-trained models are transferred to various SAR downstream tasks compared with optical ones. Lastly, we propose a novel knowledge point interpretation method to reveal the benefits of the SAR pre-trained model with comprehensive and quantifiable explanations. This study is reproducible using open-source code and datasets, demonstrates generalization through extensive experiments on a variety of tasks, and is interpretable through qualitative and quantitative analyses. The codes and models are open source.

**Keywords:** SAR image interpretation; pre-trained model; transfer learning; explainable artificial intelligence

## 1. Introduction

Synthetic aperture radar (SAR), as an active sensor, is different from optical remote sensing technology that can perform in all-day all-weather conditions. With the fast development of sensors and platforms, a large number of SAR images can be obtained day and night. In the past few years, an increasing number of researchers have studied advanced deep neural networks to solve SAR image interpretation tasks that achieve high levels of accuracy and speed [1–6].

In the realm of artificial intelligence, there has been a notable surge in attention towards pre-trained models in recent times. In the natural language processing (NLP) and computer vision (CV) fields, the fast development of pre-trained models with billions of parameters has attracted much attention [7,8]. Due to the high requirements for computational resources for training and deployment, it remains challenging for most researchers to apply the pre-trained models to downstream tasks. Since the optical remote sensing images are similar to those in computer vision, some advanced pre-trained model technologies can be easily transferred to the optical remote sensing field. Recently, a couple of works [9–11] have proposed some pre-trained models (centered on the vision transformer architecture) for optical remote sensing images by supervised or self-supervised pre-training, such as masked auto-encoders (MAE) [12]. It has been demonstrated that they excel at downstream optical remote sensing tasks [9,10].

However, the existing large pre-trained models of natural images or optical remote sensing images have a limited ability in most SAR applications. The first factor is the size of the dataset. Compared with optical remote sensing images, the current SAR dataset is small

in size due to the difficulty in data acquisition and annotation [13]. It can achieve good performance by fine-tuning the pre-trained model on an optical remote sensing dataset, but would fail with severe overfitting on SAR tasks, especially with limited training data. The second reason is the specific image characteristics. It is well known that SAR images have a different appearance from optical ones. They vary dramatically with different sensor characteristics and observation conditions. Essentially, the potential shortcomings of the optical model used to represent the SAR image should be explained comprehensively.

Our previous work discussed the limited transferability of optical data pre-trained models on SAR tasks, especially the high-level layers [14,15]. Transitive transfer learning can alleviate the domain gap to some extent. With about 80,000 high-resolution SAR image patches for scene classification, a SAR pre-trained ResNet-18 model was obtained and achieved good performance on a SAR target recognition downstream task [15]. In the SAR community, however, reusing the popular ImageNet pre-trained models is still the mainstream for various SAR image interpretation tasks [16]. Although many post hoc explanation methods have been proposed [17,18] to intuitively indicate the decision clues of deep features, they may not be fully comprehensible for SAR images [19]. In addition, the inherent weakness of optical pre-trained models transferring to SAR image tasks has not been discussed in depth with solid evidence. Thus, it is necessary to propose SAR pre-trained models and find a more comprehensible explanation method for SAR images.

To address the above-mentioned issues, in this paper, we conduct a study in the following aspects: pre-training, fine-tuning, and explaining, as shown in Figure 1. Since there still exist difficulties in SAR image classification for current open-source large-scale datasets with different satellites and resolutions, we propose an optimization method to obtain better performance in SAR scene classification to achieve pre-training. Second, both convolutional neural networks and transformer-based deep models are obtained and fine-tuned for various SAR downstream tasks, compared with optical pre-trained models. In addition, we propose an explanation method inspired by the description of knowledge points [20], in which several CAM-based methods are applied to demonstrate the benefits of SAR pre-trained models from the two perspectives of upstream and downstream tasks.

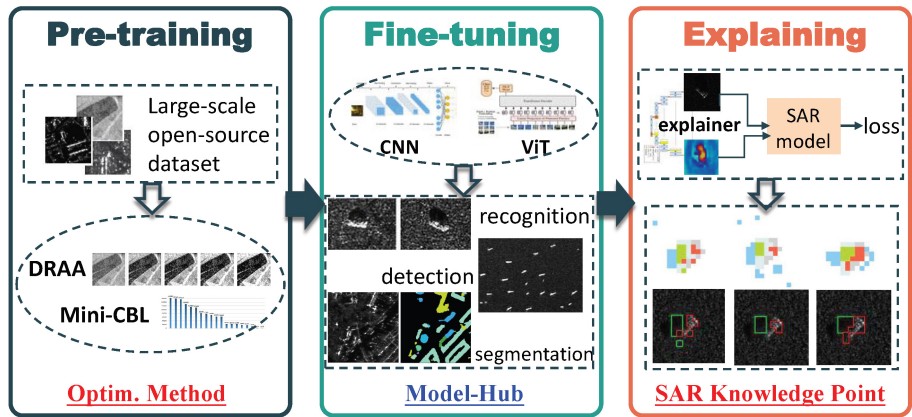

**Figure 1.** SAR-HUB overview.

The main contributions are summarized as follows.

1.  To address the challenges of data distribution drift and class imbalance problems in different SAR scene image classification datasets, we propose dynamic range adaptive enhancement (DRAE) and mini-batch class-balanced loss (Mini-CBL), which improve the model performance and feature generalization ability.
2.  Motivated by the explanation of knowledge distillation [20], we propose a novel explanation method that quantifies the knowledge of a pre-trained model transferring

to SAR downstream tasks. The results can comprehensively demonstrate the benefits of SAR-pre-trained models compared with optical ones.

3. We contribute this project to the SAR community with reproducibility (open-source code and datasets), generalization (sufficient experiments on different tasks), and explainability (qualitative and quantitative interpretations), available on 15 June 2023 via https://github.com/XAI4SAR/SAR-HUB.

The rest of this paper is organized as follows. Section 2 reviews the related work. The proposed method and the experimental results with discussions are demonstrated in Sections 3 and 4, respectively. Finally, Section 5 gives the conclusions and future perspectives.

## 2. Related Work

### 2.1. Pre-Trained Model in Remote Sensing

Recently, in the remote sensing field, a growing body of research has been devoted to pre-trained models with the goal of producing models that perform well on downstream tasks. Wang et al. [11] trained multiple pre-trained models on a large set of labeled data (i.e., Million-AID [21]). Sun et al. [9] and Wang et al. [10] obtained vision transformer-based pre-trained models using MAE [12] in order to propose customized large vision models for optical remote sensing tasks. Cha et al. [22] established the impact of augmenting the model's scale from the million scale to the billion scale and developed the first billion-scale pre-trained model in the RS domain.

Existing open-code pre-trained models in remote sensing have the ability to achieve superior performance in optical remote sensing tasks, but they are centered on optical remote sensing. Due to the specific image characteristics, there is still a dearth of pre-trained models in the SAR domain. It is essential to discuss the possible weaknesses of optical pre-trained models acting on SAR images with solid evidence and develop more sensor-specific pre-trained backbones open to the public. Previously, a simple ResNet-18 model trained on SAR scene classification was proposed and achieved better performance on SAR downstream tasks than an optical data pre-trained model [15]. This drives us to further explore the potential of the pre-trained model for SAR and explain its superiority comprehensively.

### 2.2. Post Hoc Explanation

Post hoc explanation aims to uncover the decision strategy of a trained model. The Class Activation Mapping (CAM)-based explanations provide a visual attention map where the most relevant regions to the specific class are highlighted. CAM [17] replaces the fully connected layer with a global average pooling layer and retrains it to obtain the weights of class-specific features. Successive studies have introduced gradients of categories without architectural changes or re-training, such as Grad-CAM [23] and Grad-CAM++ [24]. The related studies also include LayerCAM [18] and Score-CAM [25].

CAM-based visual explanation is intuitive but still relies on the subjective perception of humans. Sometimes, the visual interpretation of SAR images is still challenging for non-experts, and the CAM-based methods cannot provide readily comprehensible explanations. A recent work was proposed to quantify knowledge points encoded in the intermediate layers of DNNs with defined metrics to explain knowledge distillation [20]. This motivates us to develop a novel explainer to demonstrate the benefits of SAR-pre-trained models compared with optical ones.

## 3. Method

Facing the challenges of data distribution drift and class imbalance problems in various SAR scene image classification datasets that can influence the model's performance, a novel dynamic range adaptive enhancement method and a mini-batch class-balanced loss are proposed in Section 3.1 and Section 3.2, respectively. Section 3.3 presents a novel knowledge point explainer that quantifies the knowledge a pre-trained model transferred to SAR downstream tasks, which aims to reveal the superiority of SAR pre-trained models in a comprehensive manner.

### 3.1. Dynamic Range Adaptive Enhancement

In various SAR datasets, SAR image data are stored in different formats, such as the floating-point data format to record the backscattering coefficients ($\sigma_0$) in dB [26], unsigned integer numbers stored with 16 bits for quantized digital numbers (DN) with high dynamic range [27], or 8 bits for better visualization [28]. Consequently, the normalized SAR images appear differently in terms of brightness and contrast, which leads to data distribution drift in different datasets. As shown in Figure 2a, the image mean value histograms for the TerraSAR-X [27], BigEarthNet-S1 [26], and OpenSARUrban [28] datasets vary considerably from each other. Therefore, we propose dynamic range adaptive enhancement (DRAE) to adapt the model to the differences by subjecting input images to varying degrees of grayscale variation throughout training.

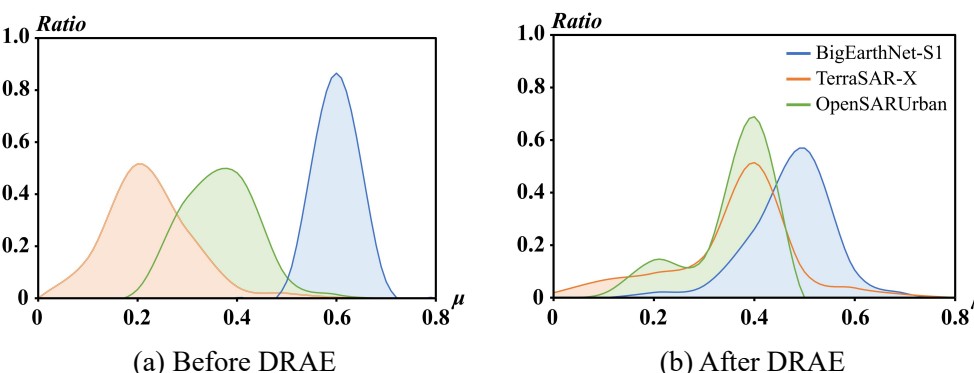

(a) Before DRAE             (b) After DRAE

**Figure 2.** SAR image mean value distribution for TerraSAR-X [27], BigEarthNet-S1 [26], and OpenSARUrban [28] datasets before and after DRAE transformation. The x-axis and y-axis represent the mean value of the normalized images, which indicates that the image's pixel values are scaled to 0–1, and the proportion in the entire dataset with different mean values.

Suppose that there are $n$ SAR image datasets with different formats, represented as $\mathbf{D} = \{D_1, D_2, \ldots, D_n\}$. The means of these normalized datasets are denoted as $\mathbf{M_D} = \{\mu_{D_1}, \mu_{D_2}, \ldots, \mu_{D_n}\}$. In SAR image processing, dynamic range reduction (DRR) based on tone mapping techniques for optical images is often applied to change the brightness and contrast, which improves visualization [29]. We denote the DRR method applied on $\mathbf{D}$ as $f_{\mathbf{\Theta}}(\cdot) = \{f_{\theta_1}(\cdot), f_{\theta_2}(\cdot), \ldots, f_{\theta_n}(\cdot)\}$, where $\theta_i$ is the hyper-parameter in the DRR method. The proposed DRAE aims to narrow the gap between min $\mathbf{M_D}$ and max $\mathbf{M_D}$ via implementing adaptive DRR with controllable hyper-parameters $\mathbf{\Theta}$.

To this end, we propose to sample the hyper-parameter $\theta_i$ from a uniform distribution $U[\theta_{i_{min}}, \theta_{i_{max}}]$ during online data augmentation, in order to ensure that the mean of an arbitrary dataset $\mu_{f_{\theta_i}(D_i)}$ lies within a proper range $[\mu_{min}, \mu_{max}]$. In this case, the disparity in averages between datasets with distinct data formats diminishes and the grayscale distributions for different datasets are closer to each other, i.e., domain drift is improved, as demonstrated in Figure 2b.

We can empirically prove that the range $[\mu_{min}, \mu_{max}]$ is around the expectation of $\mathbf{M_D}$, i.e., $\forall i, \mu_{f_{\theta_i}(D_i)} \in [\mu_{min}, \mu_{max}]$, where

$$
\begin{aligned}
\mu_{min} &= (1 - \lambda) \frac{\|\mathbf{M_D}\|_1}{n}, \\
\mu_{max} &= (1 + \lambda) \frac{\|\mathbf{M_D}\|_1}{n}.
\end{aligned}
\tag{1}
$$

$\lambda$ is a controllable hyper-parameter that determines the adjusted range. We set $\lambda$ to 0.1 in this paper.

Then, the adjustment range of hyper-parameters $\boldsymbol{\Theta}$ can be determined by optimizing the following objective:

$$\arg\min_{\boldsymbol{\Theta}_{min}}\|\mathbf{M}_{f_{\boldsymbol{\Theta}}(\mathbf{D})} - \mu_{min}\|_2,$$
$$\arg\min_{\boldsymbol{\Theta}_{max}}\|\mathbf{M}_{f_{\boldsymbol{\Theta}}(\mathbf{D})} - \mu_{max}\|_2. \tag{2}$$

Specifically, we finish the optimization by grid searching to obtain $\boldsymbol{\Theta}_{min}$ and $\boldsymbol{\Theta}_{max}$. Thus, the mean values of the datasets after data augmentation fall within $[\mu_{min}, \mu_{max}]$.

In this paper, we apply two DRR methods, Reinhard–Devlin [29] and Percentage Truncated Linear Stretch (PTLS), as $f_{\boldsymbol{\Theta}}$. Reinhard–Devlin [29] is applied on the TerraSAR-X dataset [27] with an adjustable hyper-parameter $t$:

$$f_b(x) = \frac{x}{x + (t \cdot L_a)^m}, t > 0 \tag{3}$$

where $x$ is the input SAR image. $L_a$ is the light adaptation term, defined as

$$L_a = l \cdot x + (1 - l) \cdot x_{avg} \tag{4}$$

where $l$ controls the contrast and is set to $10^{-4}$ empirically in this paper. $m$ in Equation (3) is a constant calculated from

$$m = 0.3 + 0.7\left(\frac{1 - x_{avg}}{1 - x_{min}}\right)^{1.4}, \tag{5}$$

where $x_{avg}$ and $x_{min}$ denote the average and minimum value of $x$.

PTLS is applied on the BigEarthNet-S1 [26] and OpenSARUrban [28] datasets with an adjustable hyper-parameter $v$:

$$f_v(x) = \frac{x - \text{vec}(x)^{\downarrow}[i_{min}]}{\text{vec}(x)^{\downarrow}[i_{max}] - \text{vec}(x)^{\downarrow}[i_{min}]} \tag{6}$$

where $x$ is the input SAR image. $\text{vec}(x)^{\downarrow}$ represents the vectorization of $x$ in descending order. $i_{min}$ and $i_{max}$ denote the bit order of the minimum and maximum quantiles, respectively, defined as

$$\begin{aligned} i_{min} &= v \cdot (N + 1), \\ i_{max} &= (1 - v) \cdot (N + 1), \\ 0 &< v < 1 \end{aligned} \tag{7}$$

where $N$ represents the quantity of pixels in image $x$.

The hyper-parameters $t$ in Reinhard–Devlin [29] and $v$ in PTLS are randomly sampled in online data augmentation during training. The controllable ranges of them are given in Section 4.2.2.

### 3.2. Mini-Batch Class-Balanced Loss

As depicted in Figure 3, the existing SAR image classification datasets show a substantial class imbalance. Compared with some typical class imbalance tasks in computer vision, SAR image classification datasets entail various challenges, such as fewer categories, more samples per class, and a lesser long tail effect. The current advanced loss functions addressing the class imbalance problem may not be suitable for SAR. Thus, we propose mini-batch class-balanced loss (Mini-CBL), motivated by the literature [30], to solve the specific class imbalance issue in the current SAR image datasets.

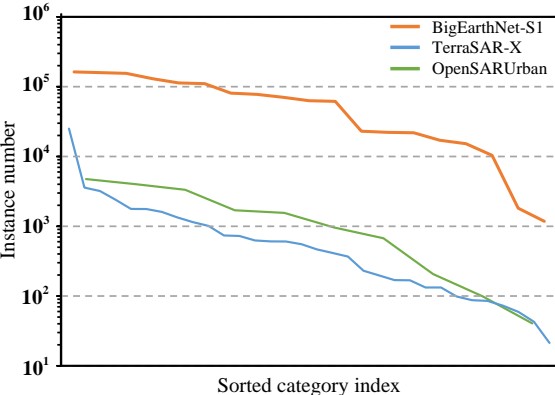

**Figure 3.** Sorted number of instances for categories in TerraSAR-X [27], BigEarthNet-S1 [26], and OpenSARUrban [28] datasets. The x-axis displays the category index sorted by the quantity of data.

The primitive method [30] defines the effective number of samples to re-balance the loss, and the global long tail label distribution is used to calculate the effective number. In a lesser long tail scenario with more samples per class, as shown in Figure 3, the re-weighting becomes insignificant. The proposed Mini-CBL for SAR datasets considers the local class imbalance in a mini-batch to solve this problem, where each sample is re-weighted based on the label distribution in a mini-batch to obtain the classification loss. The loss weight for each sample $x_i$ with label $y$ is given by

$$w_i = \frac{1 - \beta}{1 - \beta^{n_y}}, \tag{8}$$

where $n_y$ denotes the number of samples for category $y$ in a mini-batch, and $\beta$ is a hyperparameter to determine the effective number of samples, which is set to 0.995 empirically.

To prevent gradient explosion, the minimal value of $n_y$ is set to 1. Then, the defined weight $w_i$ can be used to re-balance the sample loss for every iteration with $n$ samples in a mini-batch:

$$\text{Mini} - \text{CBL} = \frac{1}{n} \sum_i w_i \cdot L(z_i, y_i) \tag{9}$$

The focal loss [31] is applied to accomplish $L(z, y)$ in subsequent experiments.

### 3.3. Knowledge Point Explainer

Regarding the convolution and pooling processing of deep neural networks as a layer-wise process of discarding information, the knowledge point was defined as an input unit where the information is much more preserved than in others [18,23,24]. It was proposed to explain knowledge distillation. Motivated by the literature [20], we design a lightweight U-Net-based explainer that generates quantified knowledge points to reveal how the prepared downstream SAR model encodes SAR images when transferring from different pre-trained models.

As shown in Figure 4, the SAR model to be explained (denoted as $f(\cdot)$) is frozen during explaining, and the U-Net model (denoted as $g(\cdot)$) is independent as it outputs the information discarding degree $\sigma$. $\sigma$ is an $N \times N$ matrix where each $\sigma_i$ denotes the degree of information discarding in the $i$th unit of the input image $x$. Consequently, the perturbed input $x'$ can be obtained by $x' \sim \mathcal{N}(x, \sigma^2)$. For implementation, it is re-parameterized by $x' = x + \Delta x$, where $\Delta x \sim \mathcal{N}(0, \sigma^2)$.

The U-Net model $g(\cdot)$ is optimized to generate a $\sigma$ discarding information as much as possible for an input $x$, and meanwhile to constrain the feature representation $f(x')$ of

the perturbed input $x'$ to remain unchanged as much as $f(x)$. To this end, the objective function can be written as

$$\arg\min_{\sigma} \frac{\mathbb{E}\,||f(x) - f(x')||^2}{\text{Var}[f(x)]} - \lambda \cdot \mathbb{E}_{x'} \sum_{i=1}^{N \times N} (\log \sigma_i + C), \qquad (10)$$

where $\mathbb{E}$ denotes expectation, and $\frac{\mathbb{E}||f(x)-f(x')||^2}{\text{Var}[f(x)]}$ measures the normalized feature perturbation. $\sum_{i=1}^{N \times N}(\log \sigma_i + C)$ quantifies the total information discarding for an image, where $C$ is a constant $\frac{1}{2}\log(2\pi e)$ to constrain the information entropy to be positive.

With the optimized $\sigma$, we can quantify different types of knowledge points. Taking SAR target recognition as the prepared downstream SAR model, the input image is first segmented into target, shadow, and clutter areas. Target scattering and shadow information are both crucial for recognition, while the clutter is denoted as background. If $\sigma_i$ has lower entropy, the SAR model will encode more discriminative information of the $i$-th unit in the input. Moreover, the units $i$ in the target and shadow areas are expected to have lower entropy than those in the clutter areas.

To this end, we set a threshold $b$ to determine the knowledge points. The average entropy of all clutter units is calculated as a baseline $\bar{H}$. If the entropy of a unit $i$ is much lower than the baseline, i.e., $\log \sigma_i + C < \bar{H} - b$, then it can be regarded as a significant knowledge point. Finally, the quantified knowledge points are visualized in different colors for target, shadow, and clutter areas, respectively, as shown in Figure 4.

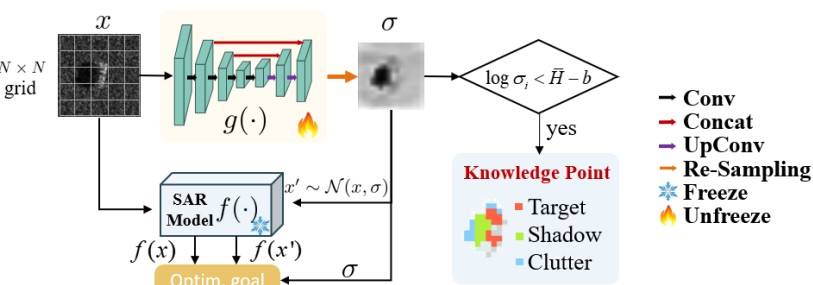

**Figure 4.** The proposed knowledge point explanation method.

## 4. Experiments

In this section, we first briefly introduce the backbone models used (Section 4.1) and then present the SAR scene classification results to demonstrate the effectiveness of the proposed DRAE and Mini-CBL methods (Section 4.2). Next, we investigate the transfer performance of SAR pre-trained models compared to ImageNet 1K [32] (IMG) pre-trained models and optical remote sensing (OPT) pre-trained models in several SAR downstream tasks (Section 4.3). Ultimately, the superiority of SAR pre-trained models is validated in Section IV-D, and experiments show that the proposed SAR knowledge point method outperforms other CAM-based methods in explaining SAR images.

### 4.1. Backbone Models

In our previous work [14,15], transitive transfer learning was proposed to refine the feature extraction ability from natural images to optical remote sensing images and to SAR images, which significantly improved the SAR image classification performance. We follow this learning pipeline in pre-training with the proposed DRAE and Mini-CBL methods.

The backbone models included six convolutional neural networks (CNNs) and two vision transformers (ViT), i.e., the ResNet series (ResNet-18, 50, 101) [33], DenseNet-121 [34], SENet-50 [35], MobileNetV3 [36], Swin-Transformer (Swin-T), and Swin-Base [37] (Swin-B). We applied the weights of ResNet-50 and Swin-T in [11] for initialization. The other backbones were trained with the NWPU-RESISC45 dataset [38] to obtain the initial weights.

All initial backbone models can be downloaded at https://github.com/XAI4SAR/SAR-HUB since 15 June 2023.

*4.2. Pre-Training: Upstream Task of SAR Scene Classification*

4.2.1. Datasets

Three SAR image datasets, TerraSAR-X [27], BigEarthNet-S1 [26], and OpenSARUrban [28], are used for pre-training. The effectiveness of the proposed DRAE and Mini-CBL is also evaluated on them.

**TerraSAR-X.** The TerraSAR-X [27] (TSX) dataset contains 46 collections based on the scene number and geographic location, spanning over 200 regions and cities worldwide. It contains Multi-Looked Grounded Detected (MGD) images derived from the spotlight mode of TerraSAR-X between November 2007 and December 2012, with an incidence angle ranging from 20° to 50°. The TerraSAR-X band refers to the X-band, which has a wavelength of 3.1 cm. Each SAR image is sliced without overlapping at pixels of $160 \times 160$ with HH polarization and a resolution of 2.9 m, which is equivalent to $200 \times 200$ square meters in physical space. The data are annotated at three levels, with gradually refined semantic granularity. In our experiments, we used the second-level annotations with 32 categories and 46,400 SAR image patches in total.

**BigEarthNet-S1.** The BigEarthNet-S1 [26] (BEN) dataset consists of 590,326 SAR slices collected from Sentinel-1 with VV and VH polarization, each of which has a size of $120 \times 120$ and 10 m resolution. The band of Sentinel-1 is C-band with 5.54 cm wavelength. All the SAR slices are in the $\sigma$ nought data format. The dataset consists of Ground-Range-Detected (GRD) pictures obtained between June 2017 and May 2018. These images were captured using the interferometric wide (IW) swath mode and encompass more than 10 European nations, including Austria, Belgium, Finland, Ireland, Kosovo, Lithuania, Luxembourg, Portugal, Serbia, and Switzerland. The 43 multi-class labels are aggregated into 19 multi-class labels using the official codes [26]. We randomly selected 10 percent of the original dataset for evaluation—59,032 SAR image patches in total.

**OpenSARUrban.** The OpenSARUrban [28] (OSU) dataset mainly contains SAR image patches of Sentinel-1, with 20 m spatial resolution. The data are presented in various distinct formats, including the original data, the enhanced grayscale 8-bit data, the visualized data in pseudo-color, and the radiometrically calibrated data. The dataset comprises Ground-Range-Detected (GRD) images acquired throughout the period from September 2016 to May 2017. The images were acquired utilizing the interferometric wide (IW) mode, including over 18 cities in China, such as Shanghai, Beijing, and others. The VV-polarized SAR image patches of 8-bit enhanced grayscale are used in our experiments, with 10 classes and 16,679 images in total.

In the subsequent experiments, the training set, validation set, and test set account for 70%, 20%, and 10% of each dataset, respectively.

4.2.2. Experimental Setup

Following reference [39], some traditional data transform methods are applied, including resizing, flipping, and cropping, and the final input image size is $128 \times 128$ pixels. We utilize AdamW [40] and OneCycleLR [41] for optimization in pre-training. The weight decay of AdamW is set to 0.05. The learning rate is controlled by OneCycleLR with hyper-parameters of $steps\_per\_epoch = 2 * (int(train\_number/batch) * GPUs + 1)$, $epochs = 150$, $anneal\_strategy = cos$. For the CNN and transformer backbone models, the initial learning rate $lr$ and the maximum learning rate $max\_lr$ are set to $5 \times 10^{-5}$ and $2.5 \times 10^{-4}$, $5 \times 10^{-6}$ and $2.5 \times 10^{-5}$, respectively. The epoch number is 300 and the batch-size is 128. All experiments are conducted on 4 GeForce RTX 3090s.

In the training process, the hyper-parameters of the proposed DRAE methods are randomly sampled to conduct online data augmentation, with $t \sim U[3.5, 4.5]$ in Equation (3) for the TSX dataset, and $v_{\text{BEN}} \sim [0, 2]$, $v_{\text{OPS}} \sim [0, 3]$ in Equation (6) for the BEN and OPS datasets, respectively. During the inference stage, they are set to 4, 1, 1.5, respectively.

### 4.2.3. Ablation Study

Table 1 describes an ablation experiment on the proposed DRAE and Mini-CBL. When both DRAE and Mini-CBL are adapted, the top-1 accuracy of the three datasets increases by an average of 1.38%, with the OSU [28] dataset showing the greatest improvement at 1.50%. In addition, the results indicate that the top-1 accuracy can be improved by implementing only DRAE or Mini-CBL. When adapting DRAE and Mini-CBL, the average top-1 accuracy can be increased by 0.84% and 0.73%, respectively.

**Table 1.** Ablation studies of the proposed DRAE and Mini-CBL.

| Backbone Model | Methods | | Top-1 Accuracy (%) | | |
| --- | --- | --- | --- | --- | --- |
| | DRAE | Mini-CBL | TSX [27] | BEN [26] | OSU [28] |
| | ✗ | ✗ | 71.29 | 62.81 | 55.03 |
| ResNet-50 [33] | ✔ | ✗ | 71.77 | 63.87 | 56.01 |
| | ✗ | ✔ | 71.97 | 63.51 | 55.84 |
| | ✔ | ✔ | **72.47** | **64.29** | **56.53** |

### 4.2.4. Effectiveness of DRAE and Mini-CBL

We demonstrate that the proposed DRAE method can improve the model's robustness when the test images are modified with different brightness and contrast. Table 2 records the test performance of models with (w/) and without (w/o) DRAE on TSX test data, where the images are changed with the Gamma transform. Clearly, the variance is lesser when DRAE is used compared to when it is not, indicating that the model with DRAE is more robust to changes in the grayscale of SAR images.

**Table 2.** We applied the *Gamma* transform to change the brightness and contrast of TSX test data with different $\gamma$. The test results were recorded, where smaller variance indicated that the model was more robust with perturbation.

| $\gamma$ | 0.95 | 0.98 | 0.99 | 1.0 | 1.01 | 1.02 | 1.05 | Variance |
| --- | --- | --- | --- | --- | --- | --- | --- | --- |
| w/ DRAE (%) | 72.51 | 72.59 | 72.61 | 72.47 | 72.54 | 72.21 | 72.34 | $1.78 \times 10^{-6}$ |
| w/o DRAE (%) | 72.14 | 72.17 | 71.89 | 71.77 | 71.54 | 71.31 | 71.33 | $1.08 \times 10^{-5}$ |

Figure 5 displays the feature maps of two ResNet-50 [33] models trained with traditional data normalization (ResNet-50-Norm) and the proposed DRAE (ResNet-50-DRAE), respectively. Three SAR image patches from TSX, BEN, and OSU are fed into ResNet-50-Norm and ResNet-50-DRAE to generate the feature maps for layer textitBN3. Due to the various data formats, their brightness and contrast vary considerably. ResNet-50-DRAE is able to represent more discriminative features for SAR images with domain drift than ResNet-50-Norm; thus, the proposed DRAE enhances the generalizability of the model.

Table 3 illustrates the effectiveness of the proposed Mini-CBL compared with the original method CBL [30], and the best performance is highlighted. *Scratch* and *Init.* in brackets indicate the model training performed from scratch and following the learning pipeline in our previous work [14,15], respectively. From the given results, Mini-CBL can obtain higher accuracy than CBL in most cases, and the gain further improves as the number of model parameters increases, e.g., a 0.05% improvement for MobileNetV3 [36] and 1.83% for Swin-B [37].

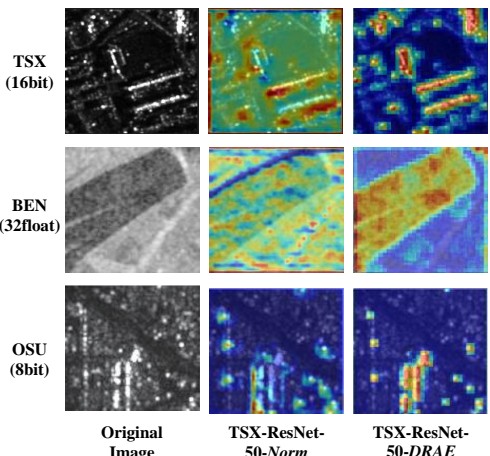

**Figure 5.** Three SAR image patches selected from TSX, BEN, and OSU are input to ResNet-50-Norm (ResNet-50 trained with conventional data normalization) and ResNet-50-DRAE (ResNet-50 trained with the proposed DRAE) to obtain the feature map in layer *BN3*.

**Table 3.** The top-1 accuracy of different backbone models for TSX [27], BEN [26], and OSU [28] datasets. *Scratch* and *Init.* indicate the model training performed from scratch and following the learning pipeline in our previous work [14,15], respectively. The best performance is in bold.

| Backbone Model | Optim. | Top-1 Accuracy(%) | | |
|---|---|---|---|---|
| | | TSX [27] | BEN [26] | OSU [28] |
| ResNet-18 [33] | Mini-CBL (Scratch) | 66.70 | 55.91 | 47.17 |
| | CBL (Init.) | 71.59 | 61.97 | 52.94 |
| | Mini-CBL (Init.) | **71.69** | **62.63** | **53.67** |
| ResNet-50 [33] | Mini-CBL (Scratch) | 67.69 | 56.08 | 45.55 |
| | CBL (Init.) | 72.31 | 63.89 | 56.04 |
| | Mini-CBL (Init.) | **72.47** | **64.29** | **56.53** |
| ResNet-101 [33] | Mini-CBL (Scratch) | 66.70 | 56.71 | 45.47 |
| | CBL (Init.) | 67.19 | **59.87** | 50.21 |
| | Mini-CBL (Init.) | **67.28** | 59.37 | **50.78** |
| SENet-50 [35] | Mini-CBL (Scratch) | 70.94 | 61.77 | 49.04 |
| | CBL (Init.) | 71.41 | 63.11 | 51.90 |
| | Mini-CBL (Init.) | **71.77** | **63.79** | **52.64** |
| MobileNetV3 [36] | Mini-CBL (Scratch) | 65.79 | 57.03 | 44.13 |
| | CBL (Init.) | 72.64 | 60.07 | 53.78 |
| | Mini-CBL (Init.) | **72.69** | **61.60** | **54.96** |
| DenseNet-121 [34] | Mini-CBL (Scratch) | 70.15 | 57.72 | 53.26 |
| | CBL (Init.) | 72.77 | 61.99 | 53.68 |
| | Mini-CBL (Init.) | **73.10** | **62.71** | **53.78** |
| Swin-T [37] | Mini-CBL (Scratch) | 68.41 | 58.87 | 44.04 |
| | CBL (Init.) | 77.87 | 66.78 | 59.31 |
| | Mini-CBL (Init.) | **78.43** | **66.74** | **59.95** |
| Swin-B [37] | Mini-CBL (Scratch) | 74.19 | 60.59 | 46.41 |
| | CBL (Init.) | 79.71 | 67.09 | 59.99 |
| | Mini-CBL (Init.) | **81.54** | **67.14** | **60.54** |

*4.3. Fine-Tuning: Downstream Tasks*

Fine-tuning is considered on different SAR downstream tasks. Table 4 gives the hyper-parameter settings.

**Table 4.** Experimental settings on different downstream tasks.

| | Target Recognition | | Object Detection | Semantic Segmentation |
|---|---|---|---|---|
| | CNNs | ViTs | | |
| Batch size | 32 | 32 | 4 | 2 |
| Optimizer | SGD | AdamW | SGD | SGD |
| Initialized learning rate | 0.01 | 0.0005 | 0.005 | 0.001 |
| Learning rate decay | StepLR | StepLR | Linear warm up | Poly schedule |
| Momentum | 0.9 | - | 0.9 | 0.9 |
| Weight decay | 0.0001 | 0.0001 | 0.001 | 0.0001 |
| Epochs | 500 | 500 | 36 | 20,000 (*max_iters*) |

1-5

4.3.1. Target Recognition

We use the MSTAR [42], OpenSARShip [13], and FuSARShip [43] datasets to evaluate the SAR pre-trained models in the SAR target recognition downstream task.

**MSTAR.** In the standard operation condition (SOC), the training set (with a depression angle of 17°) and test set (with a depression angle of 15°) consist of 10 categories of vehicles with 2747 slices and 2456 slices, respectively. We uniformly sample the training set to obtain a subset with 10% and 30% samples, with a slice count of 268 and 817, respectively. The experiments are conducted on both subsets and the original dataset.

**OpenSARShip.** There are 11,346 ship slices obtained from 41 original images in the OpenSARShip [13] dataset. We select the 8-bit grayscale images for experiments. Due to the large range of ship sizes, we divide the dataset into Small Scale, Medium Scale, and Large Scale, based on the side length within 20–60 pixels, 60–120 pixels, and greater than 120 pixels, respectively. The training and test sets are divided with a 1:1 ratio. The details are shown in Table 5.

**Table 5.** The details of OpenSARShip [13] dataset in our experiments.

| Scale | Class | Class Name | Instance No. |
|---|---|---|---|
| Small | 14 | Cargo, Dredging, Fishing, Craft, Passenger, Pilot Vessel, Pleasure Craft, Port Tender, Search Vessel, Tanker, Law Enforcement, Towing, Tug, Wing | 5522 |
| Medium | 13 | Cargo, Dredging, Fishing, Craft, Law Enforcement, Passenger, Pilot Vessel, Pleasure Craft, , Tug, Port Tender, Search Vessel, Tanker, Towing | 4813 |
| Large | 7 | Cargo, Dredging, Fishing, Passenger, Tanker, Towing, Search Vessel | 2188 |

**FuSARShip.** There are 16,144 SAR images in the dataset of 15 categories, including 6252 high-resolution images and thousands of interference samples. The dataset for our experiments comprises six categories, namely Cargo, Dredger, Fishing, Passenger, Tanker, and Tug, with a total of 3340 slices.

Instead of SGD, we use AdamW [40] to optimize the Swins [37] due to their numerous parameters. The slices in the MSTAR and FuSARShip [43] datasets are cropped and resized to $128 \times 128$, while those in OpenSARShip [13] are resized to $32 \times 32$, $96 \times 96$, and $128 \times 128$ according to the scale. In the fine-tuning procedure, all parameters in the pre-trained model

participate in iterative updates. Each task is trained 3 times and the average accuracy is taken as the final result, which is shown in Table 6, with the best performance highlighted in boldface. IMG, OPT, TSX, BEN, and OSU represent the pre-trained model with ImageNet, optical remote sensing images, and SAR images from the TSX, BEN, and OSU datasets, respectively.

**Table 6.** SAR target recognition results with different pre-trained backbones. The best performance is in bold.

| Backbone Model | | MSTAR (%) | | | OpenSARShip (%) | | | FuSARShip (%) |
|---|---|---|---|---|---|---|---|---|
| | | 10% Train | 30% Train | 100% Train | Small Scale | Medium Scale | Large Sacle | |
| ResNet-18 [33] | IMG | 83.55 | 93.67 | 97.96 | 64.47 | 81.13 | 90.31 | 71.38 |
| | OPT | 79.46 | 95.83 | 98.76 | 64.07 | 80.76 | 91.32 | 70.82 |
| | TSX | 85.94 | 96.95 | 99.88 | 64.78 | 81.21 | 91.04 | **72.86** |
| | BEN | **89.36** | 97.73 | 99.75 | **65.19** | **82.63** | **91.41** | 71.66 |
| | OSU | 87.51 | **97.98** | **99.92** | 64.85 | 81.55 | 90.40 | 71.42 |
| ResNet-50 [33] | IMG | 84.03 | 95.87 | 98.46 | 64.01 | 81.26 | 88.94 | 70.64 |
| | OPT | 85.15 | 95.83 | 98.88 | 63.67 | 80.42 | 88.76 | 70.34 |
| | TSX | 88.91 | 97.94 | 99.71 | 64.61 | 81.41 | 89.95 | **72.68** |
| | BEN | 90.31 | **98.35** | 99.59 | 64.14 | 81.26 | **90.68** | 72.20 |
| | OSU | **90.85** | 97.57 | **99.75** | **64.89** | **81.63** | 90.49 | 71.76 |
| ResNet-101 [33] | IMG | 82.97 | 94.52 | 99.75 | 64.29 | 80.80 | 89.76 | 66.37 |
| | OPT | 84.08 | 93.44 | 99.34 | 63.06 | 80.42 | 90.68 | 66.19 |
| | TSX | 80.54 | 94.97 | 99.67 | 64.32 | 80.76 | **91.22** | 66.49 |
| | BEN | **85.03** | 96.54 | **99.79** | **64.56** | **81.01** | 90.41 | 66.76 |
| | OSU | 83.18 | **96.91** | 99.63 | 64.03 | 80.96 | 91.06 | **67.15** |
| SENet-50 [35] | IMG | 84.54 | 96.54 | 99.75 | **64.72** | 80.01 | 89.31 | 69.01 |
| | OPT | 82.10 | 92.78 | 99.38 | 63.11 | 80.76 | 89.40 | 70.28 |
| | TSX | 79.05 | 92.54 | 99.67 | 63.78 | 80.46 | **91.04** | 71.00 |
| | BEN | **86.14** | **96.87** | **99.84** | 63.53 | 80.88 | 90.68 | 69.86 |
| | OSU | 82.64 | 95.46 | 99.67 | 64.25 | **81.05** | 90.22 | **72.02** |
| MobileNetV3 [36] | IMG | 62.97 | 87.13 | 99.26 | 63.27 | 80.47 | 90.40 | 69.07 |
| | OPT | 73.98 | 90.60 | 99.71 | 63.20 | 80.59 | 90.13 | 69.31 |
| | TSX | 80.49 | **96.37** | 99.71 | 63.24 | 80.69 | **91.27** | 71.06 |
| | BEN | **81.65** | 95.55 | 99.88 | **63.89** | **80.96** | 90.95 | **71.48** |
| | OSU | 77.32 | 91.22 | **99.96** | 62.66 | 80.67 | 90.68 | 70.10 |
| DenseNet-121 [34] | IMG | 78.56 | 94.80 | 99.55 | 63.83 | 80.71 | 90.31 | 70.82 |
| | OPT | 78.72 | 93.57 | 99.42 | 64.25 | 80.80 | 90.40 | 71.54 |
| | TSX | 80.87 | 93.36 | 99.59 | 63.89 | 81.50 | 89.67 | 71.48 |
| | BEN | **85.11** | **95.59** | 99.59 | 64.36 | **81.55** | **90.77** | **73.65** |
| | OSU | 81.07 | 95.34 | **99.75** | **64.80** | 80.96 | 90.59 | 72.86 |
| Swin-T [37] | IMG | 62.06 | 81.03 | 97.81 | 57.37 | 65.50 | 89.49 | 64.02 |
| | OPT | 64.74 | 80.74 | 96.29 | 59.11 | 78.01 | 89.21 | 61.25 |
| | TSX | 67.51 | **86.27** | 98.06 | **61.50** | **80.34** | **90.29** | **70.25** |
| | BEN | 65.11 | 84.86 | 98.23 | 59.98 | 80.26 | 90.06 | 64.08 |
| | OSU | **70.68** | 85.70 | **99.05** | 60.96 | 80.05 | 89.95 | 69.61 |
| Swin-B [37] | IMG | 69.77 | 82.97 | 97.72 | 61.57 | 80.09 | 88.31 | 64.20 |
| | OPT | 72.49 | 87.80 | 97.69 | 58.82 | 80.01 | 88.67 | 68.59 |
| | TSX | 72.78 | 87.18 | 98.52 | 60.34 | 80.26 | 88.76 | **70.88** |
| | BEN | 73.90 | 87.22 | 98.13 | **62.73** | 80.08 | 88.85 | 69.80 |
| | OSU | **78.52** | **91.13** | **98.85** | 62.33 | **80.30** | **90.04** | 68.65 |

From the results in Table 6, it can be seen that the SAR pre-trained models (TSX, BEN, OSU) are capable of achieving better performance than the IMG and OPT pre-trained models in most cases, especially for limited training samples. For example, *ResNet-50-OSU* can obtain 90.85% accuracy when training in 10% of the MSTAR training set, which is 5.70% and 6.82% higher than the result for the *ResNet-50-OPT* model and the *ResNet-50-IMG* model, respectively. In experiments on the OpenSARShip medium scale and the FuSARShip dataset, the SAR models can achieve average accuracy of 80.94% and 70.41%,

respectively, which is 1.46% and 2.04% better than that of the optical models. In general, the results presented in Table 6 validate the effectiveness of SAR pre-trained models. To a certain extent, they can mitigate the influence of the model generalization performance degradation induced by domain differences between optical and SAR images.

We notice that the transformer-based backbones Swin-T and Swin-B show dramatically decreasing performance with limited training data (10% and 30% train of MSTAR experiments), due to the large volume of fine-tuning parameters. It is expected that we can apply some prompt tuning strategies to improve the SAR downstream task performance in the future.

### 4.3.2. Object Detection

Three public SAR object detection datasets, SSDD [44], HRSID [45], and LS-SSDD-v1.0 [46], are examined. The training and test splitting follows the official settings.

**SSDD.** The SSDD [44] dataset is composed of 1160 images with 2456 ships with a resolution from 1 m to 15 m, collected from the RadarSAT-2, TerraSAR-X, and Sentinel-1 satellites.

**HRSID.** The HRSID [45] dataset's image size is $800 \times 800$ pixels, with data mainly sourced from Sentinel-1B, TerraSAR-X, and TanDEM-X. It contains a total of 5604 high-resolution SAR images with 16,951 ship instances.

**LS-SSDD-v1.0.** The LS-SSDD-v1.0 [46] dataset contains 1819 slices and 6015 targets obtained from 15 SAR images. The slices in the dataset have a size of $800 \times 800$ pixels and the resolution ranges from 1 m to 15 m.

We conduct experiments based on MMDetection [47], combining feature pyramid networks (FPN) [48] and fully convolutional one stage (FCOS) [49]. The detailed settings are shown in Table 4. The evaluation metrics are $AP$ and $AP^{50}$. Due to the extremely insignificant model parameter quantity of MobileNetV3 [36], we provide the results of the other seven models only, as shown in Table 7.

**Table 7.** SAR object detection results of FCOS+FPN with different pre-trained backbones. The best performance is in bold.

| Model | | SSDD | | HRSID | | LS-SSDD-v1.0 | |
|---|---|---|---|---|---|---|---|
| | | $AP$ (%) | $AP^{50}$(%) | $AP$ (%) | $AP^{50}$(%) | $AP$(%) | $AP^{50}$(%) |
| ResNet-18 [33] | IMG | **65.51** | 94.70 | 49.32 | 71.31 | 21.25 | 64.31 |
| | OPT | 61.37 | 94.41 | 46.12 | 72.27 | 21.73 | 64.82 |
| | TSX | 63.69 | **95.08** | **60.61** | **86.77** | **22.86** | **65.79** |
| | BEN | 62.21 | 94.57 | 41.43 | 72.11 | 21.60 | 64.59 |
| | OSU | 62.11 | 94.47 | 46.48 | 67.29 | 22.01 | 63.71 |
| ResNet-50 [33] | IMG | 60.53 | 93.51 | 60.68 | 87.61 | 21.74 | 64.78 |
| | OPT | 63.81 | 94.97 | 60.71 | 87.46 | 21.90 | 65.03 |
| | TSX | 67.38 | 96.91 | 59.48 | 86.21 | 22.58 | 65.69 |
| | BEN | 64.71 | 95.68 | 60.41 | 87.17 | 20.98 | 62.52 |
| | OSU | **68.78** | **97.39** | **61.24** | **88.00** | **22.67** | **66.26** |
| ResNet-101 [33] | IMG | 61.20 | 93.47 | 55.21 | **84.81** | 22.10 | **63.52** |
| | OPT | 64.01 | 95.27 | 54.31 | 82.95 | 21.89 | 62.86 |
| | TSX | **66.18** | **96.15** | **56.71** | 84.34 | **22.82** | 63.37 |
| | BEN | 65.01 | 95.13 | 55.78 | 82.76 | 22.08 | 63.09 |
| | OSU | 64.91 | 95.78 | 56.03 | 83.71 | 22.41 | 63.08 |

**Table 7.** *Cont.*

| Model | | SSDD | | HRSID | | LS-SSDD-v1.0 | |
|---|---|---|---|---|---|---|---|
| | | $AP$ (%) | $AP^{50}$(%) | $AP$ (%) | $AP^{50}$(%) | $AP$(%) | $AP^{50}$(%) |
| SENet-50 [35] | IMG | 63.55 | 94.31 | 50.38 | 77.43 | 21.38 | 63.66 |
| | OPT | **64.65** | 94.61 | 55.95 | 81.83 | 22.80 | 64.71 |
| | TSX | 64.48 | 94.25 | 56.53 | **82.66** | **24.69** | **67.18** |
| | BEN | 63.32 | 93.41 | 42.50 | 69.52 | 23.85 | 66.07 |
| | OSU | 64.28 | **94.80** | **56.62** | 82.51 | 23.92 | 66.32 |
| DenseNet-121 [34] | IMG | 21.02 | 60.17 | 27.05 | 55.31 | 10.32 | 34.72 |
| | OPT | 22.81 | 60.78 | 27.75 | 55.83 | 10.08 | 33.59 |
| | TSX | 20.27 | 60.38 | 27.01 | 54.95 | 11.26 | **37.68** |
| | BEN | **23.18** | **62.82** | 27.04 | 54.07 | **12.59** | 36.13 |
| | OSU | 22.38 | 62.14 | **28.84** | **56.78** | 11.75 | 36.07 |
| Swin-T [37] | IMG | 61.68 | 92.73 | 53.94 | 81.01 | 23.18 | 63.62 |
| | OPT | 63.35 | 94.57 | 59.07 | 86.73 | 22.93 | 65.12 |
| | TSX | 64.55 | **96.79** | 60.45 | **87.23** | 23.14 | 64.89 |
| | BEN | **64.86** | 95.67 | 53.65 | 81.53 | **23.50** | **66.81** |
| | OSU | 63.28 | 95.75 | **60.66** | 87.19 | 23.36 | 64.94 |
| Swin-B [37] | IMG | 65.15 | 95.21 | 57.10 | 86.44 | 23.27 | 64.83 |
| | OPT | 64.28 | 95.19 | 61.45 | 87.41 | 22.76 | 64.57 |
| | TSX | **65.47** | **96.86** | 61.32 | 87.11 | 23.31 | 66.62 |
| | BEN | 63.31 | 95.58 | 60.93 | 86.49 | 23.92 | 67.31 |
| | OSU | 65.18 | 96.01 | **61.87** | **87.55** | **24.06** | **67.49** |

Table 7 provides information indicating that models pre-trained on SAR datasets can achieve superior performance to the others in most situations. The *ResNet-50-OSU* model achieve the highest *AP* of 68.78% on the SSDD [44] dataset, which is 8.25% and 4.97% higher than the *ResNet-50-OPT* model and *ResNet-50-IMG* model, respectively. On the HRSID [45] dataset, the Swin-B [37] model trained by the OpenSARUrban [28] dataset has an *AP* of 61.87%, outperforming all the other models. The SAR models obtain an average *AP* of 21.11% on the LS-SSDDv1.0 [46] dataset, which is 0.59% higher than that of the optical models. In reference [11], compared to natural image pre-training models, the optical remote sensing pre-training models show an average improvement of approximately 2% in AP in object detection, while SAR pre-training models can improve the result by an additional 1% to 2%.

### 4.3.3. Semantic Segmentation

The SpaceNet6 [50] dataset is a binary semantic segmentation dataset with categories of building and background. It contains 3401 SAR images with a resolution of 0.5 m. The training set and test set are divided in a 7:3 ratio by random sampling.

We adopt DeepLabv3 [51] under the MMSegmentation [52] framework during the experiments, and the training settings are shown in Table 4. The loss used is DiceLoss [53].

Class pixel accuracy (CPA) is a commonly used indicator for segmentation and we calculate the CPA within the building area as follows:

$$CPA_{building} = \frac{TP_{building}}{TP_{building} + FN_{building}}, \tag{11}$$

where $TP_{building}$ (true positive within building area) and $FN_{building}$ (false negative within building area) represent the number of properly and improperly classified pixels in the building area, respectively. Moreover, the mean intersection over union ($mIoU$), precision

(*P*), and recall (*R*) are typically computed from both background and target regions, and the formulas for these metrics are provided below:

$$MIoU = \frac{1}{k+1} \sum_{i=0}^{k} \frac{TP}{FN + FP + TP}, \tag{12}$$

$$Presicion = \frac{TP}{TP + FP}, \tag{13}$$

$$Recall = \frac{TP}{TP + FN}, \tag{14}$$

where *TP* (true positive) and *FN* (false negative) are the number of pixels correctly and incorrectly classified. *FN* (false negative) means that the target is misused as the pixel count of the background scene and *FP* (false positive) should be classified as a target pixel point. *k* denotes the serial number of the category.

The transfer performance of the models is shown in Table 8 and some visual semantic segmentation results are provided in Figure 6. The models pre-trained on the SAR datasets have better metrics in most cases compared to others, e.g., the ResNet-50 [33] model trained on the TerraSAR-X [27] dataset can achieve a higher $CPA_{building}$ of 55.27%, which is 22.99% and 14.81% higher than that of the *ResNet-50-IMG* and *ResNet-50-OPT* pre-trained models. Overall, the average values of *mIOU* and *R* of the SAR models can reach 55.85% and 65.80%, respectively, which is 0.89% and 2.80% better than that of the optical models. The visualization results in Figure 6 demonstrate that the *ResNet-50-SAR* model can accurately segment more building areas than the *ResNet-50-IMG* and *ResNet-50-OPT* models. Notably, the majority of the regions segmented by the *IMG* models correspond to the ground truth regions, demonstrating that the *IMG* models can only segment more prominent regions in the sample, which also explains why they have the highest *P* in most cases.

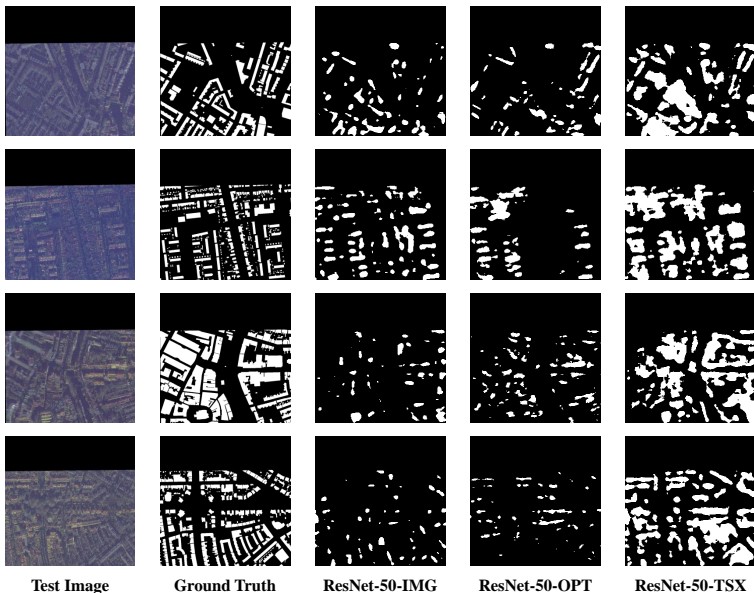

| Test Image | Ground Truth | ResNet-50-IMG | ResNet-50-OPT | ResNet-50-TSX |

**Figure 6.** The segmentation visualization results of several instances in SpaceNet6 [50] dataset.

**Table 8.** The results of semantic segmentation transfer performance verification. The best matches are highlighted in bold.

| Model | | $CPA_{building}$ (%) | $mIoU$ (%) | $P$ (%) | $R$ (%) |
|---|---|---|---|---|---|
| ResNet-18 [33] | IMG | 41.38 | 62.23 | **75.47** | 69.45 |
| | OPT | 50.08 | 61.38 | 70.32 | 72.73 |
| | TSX | **53.04** | **64.48** | 74.42 | 74.74 |
| | BEN | 52.80 | 63.85 | 73.46 | 74.49 |
| | OSU | 57.23 | 63.55 | 71.89 | **76.24** |
| ResNet-50 [33] | IMG | 32.28 | 60.63 | **79.54** | 65.47 |
| | OPT | 40.46 | 61.82 | 75.06 | 68.98 |
| | TSX | **55.27** | 60.41 | 68.11 | **74.54** |
| | BEN | 44.98 | 62.78 | 74.61 | 71.04 |
| | OSU | 50.95 | **62.79** | 72.32 | 73.46 |
| ResNet-101 [33] | IMG | 40.32 | 55.32 | **71.70** | 68.53 |
| | OPT | 21.93 | 54.60 | 68.04 | 59.83 |
| | TSX | **48.65** | 57.29 | 64.98 | **70.83** |
| | BEN | 42.78 | 55.16 | 64.04 | 63.95 |
| | OSU | 42.26 | **57.60** | 66.19 | 68.45 |
| SENet-50 [35] | IMG | 25.03 | **56.20** | **70.98** | 61.49 |
| | OPT | 28.37 | 55.82 | 67.06 | 62.57 |
| | TSX | 37.90 | 54.01 | 61.69 | 65.37 |
| | BEN | **46.10** | 53.44 | 60.92 | **68.23** |
| | OSU | 36.48 | 53.83 | 61.55 | 64.77 |
| DenseNet-121 [34] | IMG | 24.28 | 46.94 | 52.64 | 52.82 |
| | OPT | 27.53 | 49.35 | 53.21 | 56.32 |
| | TSX | **28.34** | 51.21 | 54.12 | **57.32** |
| | BEN | 26.75 | 52.01 | 55.28 | 56.30 |
| | OSU | 27.38 | **53.75** | **56.71** | 56.84 |
| Swin-T [37] | IMG | 18.53 | 52.19 | 62.24 | 57.70 |
| | OPT | 22.17 | 49.26 | 56.05 | 57.35 |
| | TSX | **40.15** | 52.94 | 60.39 | **65.73** |
| | BEN | 28.47 | **53.82** | **62.55** | 61.83 |
| | OSU | 26.85 | 52.93 | 61.23 | 60.86 |
| Swin-B [37] | IMG | 20.08 | **51.85** | **60.63** | 58.04 |
| | OPT | 22.83 | 51.70 | 59.68 | 58.90 |
| | TSX | 22.86 | 47.75 | 54.52 | 56.64 |
| | BEN | 26.17 | 50.39 | 57.50 | 59.36 |
| | OSU | **32.56** | 48.90 | 56.29 | **60.77** |

*4.4. Explanations*

4.4.1. Feature Analysis with Class-Specific Explanations

We first briefly indicate the distinction of the SAR image features that different pre-trained models represent. Three pre-trained models are selected for illustration, which are *ResNet-50-IMG*, *ResNet-50-OPT*, and *ResNet-50-BEN*. Four SAR images with annotations of *Lake*, *Channel*, *Forest*, and *Agricultural Land* are input to the three pre-trained models to obtain the corresponding feature maps of Layer 1, 2, 3, and 4 of ResNet-50, as shown in Figure 7a. The low-level features (Layer 1) of the three pre-trained models are similar, while the higher-level features differ slightly and vary considerably in the last layer. How representative are the high-level deep features of different pre-trained models for SAR images? We adopt CAM-based methods for explanation.

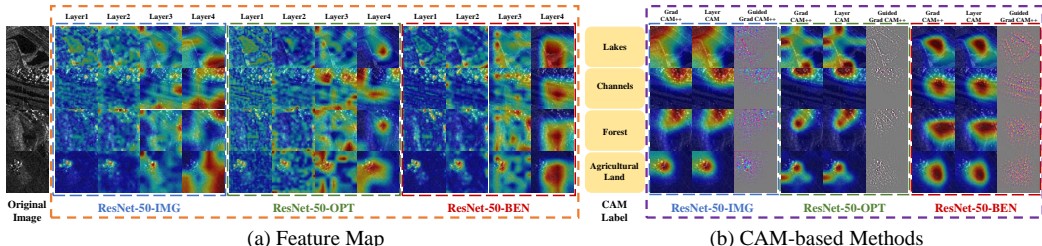

(a) Feature Map          (b) CAM-based Methods

**Figure 7.** Visualizations of CAM-based methods and the feature maps of ResNet-50 [33] models. The *ResNet-50-IMG*, *ResNet-50-OPT*, and *ResNet-50-BEN* models are fixed, except for the classification layer, and trained with the TerraSAR-X dataset for SAR image classification.

The model parameters of *ResNet-50-IMG*, *ResNet-50-OPT*, and *ResNet-50-BEN* are fixed, except for the classification layer, which is trained with the TerraSAR-X dataset for SAR image classification. Given the four test images of *Lake*, *Channel*, *Forest*, and *Agricultural Land*, the obtained models can output the probability for each class. As reviewed in Section 2.2, CAM-based explanations highlight the most relevant regions to a specific class. We try to adopt GradCAM++ [24], LayerCAM [18], and Guided GradCAM++ [24] to obtain the decision clues for each model, as shown in Figure 7b. The results reveal that *ResNet-50-IMG* and *ResNet-50-OPT* mostly concentrate on regions with strong backscattering in SAR images for decision, even when the predicted label is *Lake* or *Channel*. As a comparison, the SAR pre-trained model provides more representative high-level features related to semantics, highlighting the discriminative regions of Lake, Channel, Forest, and Agricultural Land, respectively.

Notably, the CAM-based explanations for SAR images are less comprehensible than those for optical images. For instance, Guided GradCAM++ [24] can provide high-resolution and class-discriminative feature map explanations, but the pixel-level feature visualization for SAR images is not accessible enough. Hence, we provide more interpretable results to illustrate the effectiveness of the SAR pre-trained model in the next section.

### 4.4.2. Knowledge Point Explanations

**KP Explainer Optimization.** The proposed KP explainer is optimized with Equation (10), minimizing the feature difference $Dist(f(x), f(x'))$ and maximizing the information discarding $Entropy(\sigma)$ simultaneously. In this way, the explainer can capture the most crucial information in the input that is encoded by the downstream model for prediction. Figure 8 shows the loss attenuation during explainer training for different transferred models.

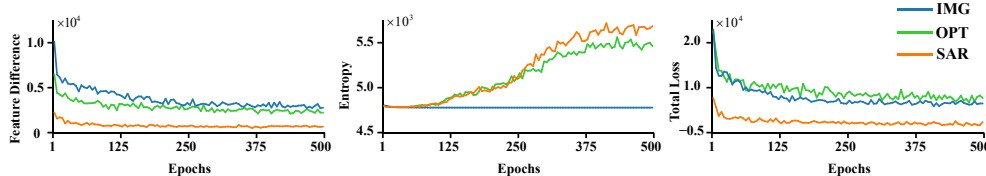

**Figure 8.** The loss attenuation during the explainer training, including the feature difference $Dist(f, f')$ ( **Left**), information discarding $Entropy(\sigma)$ ( **Middle**), and total loss ( **Right**).

As discussed before, the proposed SAR pre-trained model outperforms the counterpart ImageNet (IMG) and optical remote sensing data (OPT) pre-trained ones in the SAR target recognition downstream task. Figure 8 demonstrates that in the very beginning of explainer training, where the information discarding is almost the same, the feature difference of the SAR pre-trained downstream model is much less than that of the two optical ones, i.e., the SAR pre-trained model offers a more robust feature representation for SAR target images. After training for epochs, the SAR pre-trained model maintains smaller feature variance with less information preserved in the input than the IMG and OPT pre-trained ones.

**KP Quantity Discussion.** We illustrate the superiority of the SAR pre-trained model by discussing the quantity of KP during downstream task fine-tuning. The pre-trained model is fine-tuned for SAR target recognition for 300 epochs in total. We save the fine-tuned model every 50 epochs and train a KP explainer for each one. Thus, as the pre-trained model is being fine-tuned, the KP quantities for target ($N_{KP}^{Target}$) and shadow ($N_{KP}^{Shadow}$) are as recorded in Figure 9 on the left and right, respectively. As seen in Figure 9 on the left, the SAR pre-trained models exhibit a larger number of KPs in the target area compared to other models with the rise in epochs, which indicates that the SAR pre-trained model demonstrates a greater ability to focus its attention on the target area. In Figure 9 on the right, it can be observed that the number of KPs in shadow regions acquired by the SAR pre-trained models exceeds that of the optical models. This finding suggests that SAR models exhibit a greater emphasis on the shadow area. To our knowledge, the strong backscattering region together with the shadow information of a target are crucial for recognition. As a result, KPs in target and shadow areas can reflect the effective information encoded by a model.

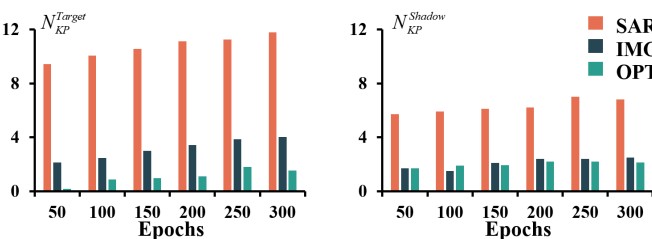

**Figure 9.** The number of KPs in target (**left**) and shadow areas (**right**) encoded during fine-tuning, transferring from ImageNet (ResNet-50-IMG), optical remote sensing data (ResNet-50-OPT), and SAR pre-trained model (ResNet-50-SAR), respectively.

It is notable that the SAR pre-trained model can learn many more KPs in the target and shadow areas than the IMG and OPT ones at the very beginning of fine-tuning (50 epochs). This explains the faster convergence and better performance of the SAR pre-trained model in transferring to SAR target recognition, which promotes the model to learn more discriminative feature representations in crucial regions (target and shadow). As a comparison, the OPT-trained one can rarely learn any valid KPs in the target scattering region. The IMG and OPT pre-trained models have comparable abilities to encode knowledge in the shadow area.

**KP Compactness Discussion.** According to the proposed method, the KP unit $i$ is determined by a pre-defined threshold $b$ and the average entropy $\bar{H}$ in clutter, i.e., $\log \sigma_i < \bar{H} - b$. We consider that a KP unit $i$ is compact if it preserves much information, i.e., $\log \sigma_i$ has a very small value and this unit remains a KP with an increasing threshold $b$. A good pre-trained model can successfully encode more compact KPs in crucial regions, such as target and shadow. As a result, we present Figure 10 to demonstrate the superiority of the SAR pre-trained model in learning more compact KPs compared to optical ones.

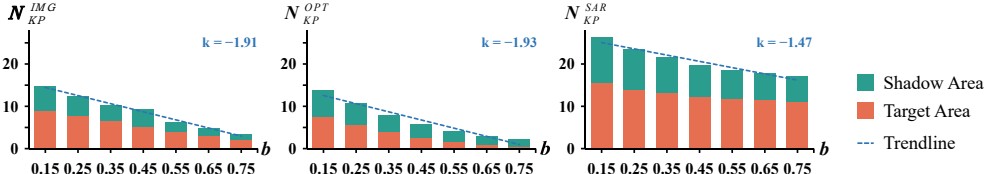

**Figure 10.** The number of KPs in target and shadow areas with different thresholds $b$. ResNet-50-IMG (**Left**), ResNet-50-OPT (**Middle**), and ResNet-50-SAR (**Right**) represent transfer from different pre-trained models. $k$ is the slope of the trendline.

In Figure 10, the number of KPs (*y*-axis) in target and shadow areas with different thresholds $b$ (*x*-axis) is plotted. We additionally draw a trendline with a fitted slope

for quantification. Obviously, the SAR pre-trained model shows remarkable advantages concerning compact KPs for SAR targets. The IMG and OPT pre-trained models appear to make no major difference in learning compact KPs when transferring to the SAR target recognition downstream task.

**KP Visualization.** Figure 11 shows the visualization of KPs on three selected MSTAR target images. For each sample, the explainer generates the information discarding in every unit and quantifies it with entropy. The KPs in target, shadow, and clutter are determined based on a pre-defined threshold *b*. Here, we test different values of *b* for visualization.

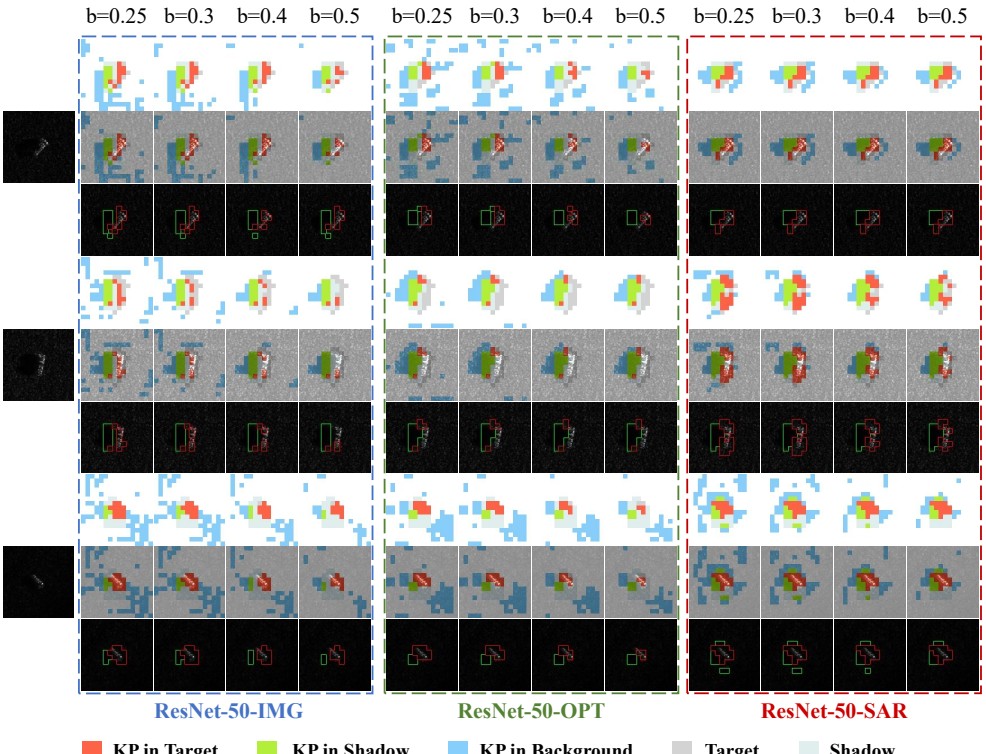

**Figure 11.** Visualization of KPs on three selected MSTAR target images. The first row visualizes the KP distribution in target (red), shadow (green), and clutter areas (blue), respectively. The second row shows that the three types of KPs are associated with particular regions within the original image. Target and shadow KP regions mapped to the original SAR images are given in the third row. The explanation results with different thresholds *b* are shown.

As the perturbation increases, the quantity of KPs within a particular region serves as a direct indicator of the degree of information loss that transpires in that area. In summary, a larger quantity of key points (KPs) indicates a reduced loss of information within the region and greater attention of the network to the region. In addition, the threshold *b* reflects the difference in information entropy between the target area and the background area. If more KPs are retained in the target area when b increases, it can be inferred that the target area possesses a greater amount of information compared to the background area. This suggests that the model exhibits a heightened capability to discriminate between the target and the background.

KPs in the target, shadow, and clutter regions are displayed in red, green, and blue, respectively, in the first and second rows. Other non-KP units are distinguished by color in the target and shadow zones. As the threshold *b* increases, the number of KPs is observed to decrease. Compared to IMG and OPT pre-trained models, the increasing threshold *b* only substantially filters out KPs in clutter, whereas the SAR pre-trained model is more robust in target and shadow areas. We apply the KP result to the original input image and circle the KP regions in the target and shadow, as shown in the third row. It is evident that the SAR pre-trained model can effectively encode more predictive information in critical regions.

## 5. Conclusions

We conduct a study of pre-training, fine-tuning, and explanation for SAR in this paper. In order to address the issues of data distribution drift and class imbalance problems in SAR scene image classification datasets, we propose an optimization method incorporating DRAE and Mini-CBL. Following this, we train a set of SAR pre-trained models using the optimization method, and a large number of experiments are conducted to demonstrate that the SAR pre-trained model outperforms the optical data pre-training model in SAR downstream tasks. In addition, we propose a knowledge point explainer to prove the benefits of SAR pre-trained models. The experimental results show that the proposed explainer can provide more comprehensive and quantifiable explanations for SAR images than traditional CAM-based methods. We hope that the SAR pre-trained models and KP explainer can contribute to research in other SAR downstream tasks and explainable tools in the community.

In the future, we will further focus on how to transfer ViTs to SAR downstream tasks more effectively, considering their subpar performance in recognition tasks. Furthermore, we shall investigate strategies to efficiently apply the explanations acquired from the KP explainer to SAR tasks, such as SAR target recognition and detection, as opposed to merely employing it as a tool.

**Author Contributions:** H.Y. and Z.H. conceived and designed the experiments; H.Y., X.K., L.L. and Y.L. performed the experiments; H.Y. and Z.H. analyzed the data; Z.H. and H.Y. wrote the paper. All authors have read and agreed to the published version of the manuscript.

**Funding:** This work was supported by the National Natural Science Foundation of China under Grant 62101459 and the China Postdoctoral Science Foundation under Grant BX2021248.

**Data Availability Statement:** The dataset and the log files can be obtained by contacting us through huangzhongling@nwpu.edu.cn, which those are welcome to do so.

**Acknowledgments:** The authors would like to thank the reviewers and the editors for their constructive comments to improve this paper's quality.

**Conflicts of Interest:** The authors declare no conflict of interest.

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
