# Peer review of "SAR-HUB: Pre-Training, Fine-Tuning, and Explaining"

_remotesensing, doi:10.3390/rs15235534_

Round 1

Reviewer 1 Report

Comments and Suggestions for Authors

In this paper, an exhaustive investigation of pre-training, fine-tuning, and explanation for SAR are presented. This paper is well written and organized. 

There are a few suggestions.

1. On Page 4 line 139, the λ set to 0.1. Please provide more details about how to set this parameter.

2. In this paper, TerraSAR-X, BigEarthNet-S1, and OpenSARUrban datasets are used. Please present more details about those three datasets.

3. Figure 6 demonstrates the segmentation visualization results of several instances. Please specify which dataset is used.

Comments on the Quality of English Language

In this paper, an exhaustive investigation of pre-training, fine-tuning, and explanation for SAR are presented. This paper is well written and organized. 

There are a few suggestions.

1. On Page 4 line 139, the λ set to 0.1. Please provide more details about how to set this parameter.

2. In this paper, TerraSAR-X, BigEarthNet-S1, and OpenSARUrban datasets are used. Please present more details about those three datasets.

3. Figure 6 demonstrates the segmentation visualization results of several instances. Please specify which dataset is used.

Reviewer 2 Report

Comments and Suggestions for Authors

This study focuses on providing methods and tools in Pre-training, Fine-tuning, and Explaining the deep neural network models for SAR image processing. Following are some suggestions from the reviewer.

The theoretical rationale of the proposed DRAE method should be better introduced. It should be better explained why the key steps and parameters are settled.

The difference between figure 9 left and right is not introduced.

The visual explanation of the proposed KP explainer should be better provided. Moreover, how the explanation helps the improvement of the model should be discussed.

The conclusion is suggested to be more specific, to highlight the key findings of the proposed study.

Reviewer 3 Report

Comments and Suggestions for Authors

The manuscript provides a clear overview of the research conducted in the realm of artificial intelligence and remote sensing, explicitly focusing on pre-trained models for SAR (Synthetic Aperture Radar) image analysis. The introduction effectively introduces the critical problem, which is the limitations of optical pre-trained models when applied to SAR image tasks due to the sensor gap between the two data types. This sets the stage for the reader to understand the research context. Objectives of research clearly outline the three main aspects of your study: Pre-training, Fine-tuning, and Explaining. The Materials and Methods section provides insight into your methodology, which includes collecting SAR scene image classification datasets, using deep neural networks (CNNs and vision transformers), and proposing new techniques to address SAR image classification challenges. It mentions that pre-trained models are transferred to SAR downstream tasks, highlighting the practical application of the research. The research also discusses a novel knowledge point interpretation method for explaining the benefits of SAR pre-trained models, which adds an exciting dimension.

Key Features: The abstract emphasizes the reproducibility of the study using open-source code and datasets, the demonstration of generalization through extensive experiments, and the interpretability of the results. These are essential aspects that demonstrate the robustness and applicability of your research.

My review:

- Explicitly state why the study is essential. What are SAR-HUB's potential consequences? This can further emphasize the significance of the research.

- Mention any limitations or scope constraints of the study. For example, if the research focuses solely on certain types of wetlands or excludes certain factors, acknowledging these limitations can provide transparency.

- Clearly define the scope of the study, including any limitations or constraints.

- Ensure that all references are cited correctly and consistently throughout the introduction.

- Offer insights into potential future research directions or unanswered questions from your study.

- Explicitly mention any limitations of your study, such as data limitations, assumptions made, or constraints in the methodology. Being transparent about limitations helps readers interpret the results accurately.

Comments on the Quality of English Language

The manuscript is generally well-structured and clear, making it easy for the reader to grasp the purpose and significance of your research. Overall, it provides a comprehensive and well-organized overview of the study and covers all the necessary components.

Reviewer 4 Report

Comments and Suggestions for Authors

The paper presents an original solution to the task of object detection using Synthetic Aperture Radar, consisting of dynamic range adaptive enhancement method and a mini-batch class-balanced loss, and then pre-trained models transferred to various SAR downstream tasks.

Comments:

1. Correct the Abstract, discarding its first half, which concerns the general description of the problem, which should be moved to the Introduction section.

2. Delete the word "innovative method" throughout the text of the paper, leaving it for use by the paper's reviewers.

3. Instead of boasting about the novelty of your work, in the introduction section, formulate the thesis of the work (demonstrating that through ..., it is possible to ..., which enables ... and brings the following results) and two or three goals of the work resulting from it, which become its chapters.

4. The Conclusions section is a summary of the work, not a presentation of the most important qualitative and, above all, quantitative conclusions from the research conducted.

Then, you should present a specific plan for further research on the topic of paper.

Reviewer 5 Report

Comments and Suggestions for Authors

Dear Authors,

I believe that the article is innovative and has great importance for the SAR community. Almost exceptionally, all the pieces of code and SW provided in the GITHUB.

Nevertheless, I have the following questions and suggestions:

1.      Please state working wavelengths of TerraSAR-X, BigEarthNet-S1, and OpenSARUrban. Please also state polarization used by TerraSAR-X. Additionally, for all datasets, it’s necessary to provide the info regarding the range of incidence angles used by all datasets as well as the time acquisition range and areas covered by images.

2.      Was Sigma nought (dB scale) used in all datasets? Which sensor modes were used for data acquisition?

3.      What kind of SAR image pre-processing was applied to the datasets? I mean operations such as Incidence angle normalization or Range-Doppler Terrain correction, for example.

4.      Please provide additional info on the ship datasets. How they were derived and how accurate are they? The concern is that there are many more classes of ships than classes in datasets. Moreover, some ships of different classes might be similar in appearance such as bulkers (cargo classes) and tankers (tanker class) (see here for, example https://www.merchantnavydecoded.com/life-on-ship-bulk-carrier-oil-tanker/ ). Obviously, bulkers have large covers over their holds but it should be explained how they might be reliable (at any angle on a ship relative to a SAR sensor and on a ship of any size within a class) distinguished from a network of pipes present on the top decks of tankers considering limitations of the sensors and noise presence. With all of that being said, it might be possible that derived models can produce overfitting. Therefore it is necessary to perform some kind of feature analysis to make a conclusion about which particular features in datasets (and their threshold levels) permit making the conclusion regarding the vessel classes classification. It is necessary that these features would have a physical meaning.

I believe, that after the questions are answered, the article would have even higher importance.

Thank you and good luck with the publishing of the article and your future research!

Your Reviewer

Round 2

Reviewer 5 Report

Comments and Suggestions for Authors

Dear Authors,

Thank you very much for addressing all the questions. You've done a great job! I recommend the article for publishing and I believe that it will be very interesting to the Readers of the Journal. I wish you great success in your future research and careers.

Best Regards,

Your Reviewer

Author Response

Dear reviewer,

We are very grateful for your kind appraisal. Thank you for your time and dedication in reviewing our manuscript. We are honored to have had the opportunity to receive feedback from someone with your level of expertise.

Best regards.

Haodong Yang.